# Genome-Wide Identification of PEBP Gene Family in *Solanum lycopersicum*

**DOI:** 10.3390/ijms24119185

**Published:** 2023-05-24

**Authors:** Yimeng Sun, Xinyi Jia, Zhenru Yang, Qingjun Fu, Huanhuan Yang, Xiangyang Xu

**Affiliations:** Laboratory of Genetic Breeding in Tomato, Key Laboratory of Biology and Genetic Improvement of Horticultural Crops (Northeast Region), Ministry of Agriculture and Rural Affairs, College of Horticulture and Landscape Architecture, Northeast Agricultural University, Mucai Street 59, Harbin 150030, China

**Keywords:** gene family analysis, PEBP gene family, plant growth and development, *Solanum lycopersicum*, tomato

## Abstract

The PEBP gene family is crucial for the growth and development of plants, the transition between vegetative and reproductive growth, the response to light, the production of florigen, and the reaction to several abiotic stressors. The PEBP gene family has been found in numerous species, but the SLPEBP gene family has not yet received a thorough bioinformatics investigation, and the members of this gene family are currently unknown. In this study, bioinformatics was used to identify 12 members of the SLPEBP gene family in tomato and localize them on the chromosomes. The physicochemical characteristics of the proteins encoded by members of the SLPEBP gene family were also examined, along with their intraspecific collinearity, gene structure, conserved motifs, and cis-acting elements. In parallel, a phylogenetic tree was built and the collinear relationships of the PEBP gene family among tomato, potato, pepper, and Arabidopsis were examined. The expression of 12 genes in different tissues and organs of tomato was analyzed using transcriptomic data. It was also hypothesized that *SLPEBP3*, *SLPEBP5*, *SLPEBP6*, *SLPEBP8*, *SLPEBP9*, and *SLPEBP10* might be related to tomato flowering and that *SLPEBP2*, *SLPEBP3*, *SLPEBP7*, and *SLPEBP11* might be related to ovary development based on the tissue-specific expression analysis of SLPEBP gene family members at five different stages during flower bud formation to fruit set. This article’s goal is to offer suggestions and research directions for further study of tomato PEBP gene family members.

## 1. Introduction

A highly conserved protein known as the phosphatidylethanolamine-binding protein (PEBP) gene family has been found in a variety of biological tissues. These proteins were initially isolated from the brains of cows and were given the name phosphatidylethanolamine-binding protein due to their strong affinity for phospholipids [1,2]. The structure of a number of discovered PEBP-like family members reveals that PEBP has a conserved domain with a sizable central β-sheet. A small β-sheet and an α-helix make up this large β-sheet [3,4].

The PEBP gene family controls a wide range of biological processes. Some studies have found that PEBP genes can be involved in regulating flowering and growth-related signaling pathways [5,6], and play an important role in plant type differentiation [7]. The PEBP gene family in plants can be divided into three distinct subgroups: the FLOWERING LOCUS T-like gene, which is thought to be involved in encoding florigen [8,9]; the TERMINAL FLOWER1 (TFL1)-like gene related to inhibiting flowering [7,10]; and the MOTHER OF FT AND TFL1 (MFT)-like gene related to regulating seed germination [11,12]. The Arabidopsis *FT*, *TSF*, and tomato *SFT* genes are FL-like genes. *SFT* has been demonstrated to produce a signal to guide flowering and play a regulatory role in plant meristem growth [13,14]. *TFL1*, *ATC*, and *BFT* in Arabidopsis and the *SP* gene of tomato are TFL1-like genes, and *SP* can antagonize the effect of *SFT* [15]. The *SFT/SP* balance controls the flowering initiation time [9], constitutes the flowering regulation mechanism together with florigen, and regulates the growth and stoppage of the branch meristem [16]. The Arabidopsis *MFT* gene is an MFT-like gene that is involved in seed germination. According to previous studies, replication events in the plant PEBP gene family occur at the same time as plant evolution. Only MFT-like genes have been found in bryophytes [12], and MFT-like genes may be traced back to the origin of terrestrial plants [17]. Due to the lack of sequenced genomes of gymnosperms, the origins of FT-like and TFL-like genes are controversial [7,18], but their gene duplication events coincide with the evolution of seed plants. Further studies suggest that FT-like and TFL-like genes have differentiated in gymnosperms and FT-like genes have recently undergone selection [17]. PEBP family members have been identified in many plants, such as tobacco [19], maize [20], rice [21], and soybean [22,23]. Six members—*FT*, *TSF*, *TFL1*, *BFT*, *ATC*, and *MFT*—have been discovered in Arabidopsis [7,24,25]. Ten members were identified in *Brassica oleracea*, some of which affected the curd formation and flowering of cauliflower, and a study suggested that the high expression of the *TFL1* gene and the low expression level of flowering-promoting genes (*FT*, *TSF*) may be among the factors that affect curd formation in cauliflower [26]. Regarding the identification of the PEBP gene family in cotton, four cotton varieties, two tetraploid and two diploid, were identified, with 21 and 20 members identified in the former and 10 each in the latter [27].

One of the most popular vegetable crops is the tomato. The tomato fruit has a unique flavor, can be eaten raw or cooked, and has great economic and nutritional value. According to inflorescence development at the top of the stem, tomatoes have two types of growth, determinate and indeterminate, being a typical sympodial branching crop [28]. The determinate growth type is usually characterized by short plants, early fruit maturity, concentrated maturity, and relatively low yield, while the indeterminate growth type is the opposite, with tall plants and a long growth period [15,28]. Because these two types of tomato have obvious differences in many aspects, it is very important to study tomato growth habits and their regulatory genes. Eight members of the tomato PEBP gene family have been found: *SP2G*, *SP3C*, *SP3D (SFT) SP3I*, *SP5G*, *SP6A*, *SP*, and *SP9D* [29,30]. These eight genes have similar functions in the growth process of tomato and play direct and indirect roles in regulating the tomato plant type. The tomato *SP* gene, which can inhibit tomato flowering, is orthologous to the *CEN* gene and the Arabidopsis *TFL1* gene [15,31,32]. The *SP* gene can regulate the determinate or indeterminate growth of tomato. After gene mutation, tomato changed from indeterminate to determinate growth, the plant type became compact and dense, and maturity was consistent [15]. The *SP* gene mutant is beneficial to the mechanical harvesting of tomato [15]. *SP3D* is homologous to the *FT* gene of *Arabidopsis thaliana*, and its deletion can prolong the flowering period of tomato [33]. *SP5G* regulates the photoperiod sensitivity of tomato [34]. Although multiple members of the tomato PEBP gene family have been identified and researched in tomatoes, they have not been systematically analyzed by bioinformatics. Using bioinformatics techniques, 12 members of the SLPEBP gene families were examined in this study, chromosomal localization was performed, and a phylogenetic tree was constructed. In addition, their collinearity, gene structure and conserved motifs, protein three-dimensional structures, and types of cis-acting elements were analyzed. Transcriptomic information from the cultivated tomato variety Heinz was used to examine the expression of 12 members of the SLPEBP gene family in different organs and tissues. The tissue expression specificity of tomato flowers in five different periods was analyzed. The findings of this study will help further elucidate the function of the PEBP genes in tomato flowering and plant development, offering suggestions for improving current tomato varieties and creating new ones.

## 2. Results

### 2.1. Identification of PEBP Gene Family Members in Tomato and Analysis of the Physicochemical Properties of Proteins

Through HMMER 3.0 software analysis and screening with HMMER and SMART online software, 12 PEBP family members were analyzed and named *SLPEBP1-12*. Basic information about the SLPEBP gene family and the physicochemical properties of its encoded proteins is shown in Table 1. In addition, the named genes are also shown in the table.

The shortest SLPEBP gene family coding sequence length is *SLPEBP8* at 423 bp and the longest is *SLPEBP2* at 699 bp; the average length is 527 bp. SLPEBP family genes are distributed across seven tomato chromosomes. The physicochemical properties of the proteins were predicted by ExPASy online software. The smallest molecular weight was 16.06 for *SLPEBP8*, and the largest was 24.80 for *SLPEBP2*. The theoretical isoelectric point ranged from 5.26 (*SLPEBP7*) to 9.63 (*SLPEBP2*), and the instability index ranged from 28.46 (*SLPEBP10*) to 50.26 (*SLPEBP9*). The maximum aliphatic index value was 92.36 (*SLPEBP8*), and the minimum value was 72.71 (*SLPEBP3*). The results of hydrophilicity and hydrophobicity analysis indicated that all SLPEBP family proteins are hydrophilic proteins. The results of subcellular localization of the proteins analyzed using online software showed that all SLPEBP family proteins are cytoplasmic proteins.

### 2.2. Chromosome Localization and Collinearity Analysis of PEBP Family Members in Tomato

The relationship between the location of the SLPEBP gene family on the chromosomes and the collinearity within the species is shown in Figure 1. Members of the SLPEBP family are only partially distributed on the chromosomes of tomato and are scattered, with 12 genes distributed across seven tomato chromosomes, mostly relatively close to the telomeric region of the chromosome. Only *SLPEBP5* among the 12 SLPEBP family members is located away from the telomeric region. Among the 12 SLPEBP family members, only *SLPEBP5* was in a region with low gene density. The analysis of collinearity between genes demonstrated the replication relationships between genes, and these relationships within species are shown in Figure 1. SLPEBP gene family members have only one collinear relationship within species, i.e., *SLPEBP1* with *SLPEBP9*.

To determine the collinear relationships of the 12 SLPEBP family genes among other species, collinearity analysis was performed using Arabidopsis, potato, and pepper with PEBP family members from tomato. The results of the analysis are displayed in Figure 2. Tomato had 11 collinearities with potato, 10 collinearities with pepper, and 4 collinearities with Arabidopsis. *SLPEBP2* and *SLPEBP5* had no collinearity with genes from other species. Among the three species, SLPEBP is most closely related to STPEBP. Comparative analysis of the SLPEBP family with other species can provide a research reference for analyzing the genetic relationships and gene functions of species. Collinearity relationship of tomato SLPEBP gene with potato, pepper and *Arabidopsis thaliana* can be found in Appendix A.

### 2.3. Phylogenetic Relationship Analysis

To investigate the phylogenetic relationships of SLPEBP genes with other homologous genes, a phylogenetic tree was constructed using the amino acid sequences of 10 identified ATPEBP proteins, 10 STPEBP proteins, 11 CAPEBP proteins, and 12 SLPEBP proteins (Figure 3). The phylogenetic tree was divided into three subfamilies: TFL1, FT, and MFT. Sixteen genes belonged to the TFL1 subfamily (five SLPEBP genes: *SLPEBP1*, *SLPEBP2*, *SLPEBP4*, *SLPEBP9*, *SLPEBP11*). Fifteen genes belonged to the FT subfamily (four SLPEBP genes: *SLPEBP5*, *SLPEBP7*, *SLPEBP8*, *SLPEBP12*), and twelve genes belonged to the MFT subfamily (three SLPEBP genes: *SLPEBP3*, *SLPEBP6*, *SLPEBP10*).

### 2.4. Conserved Motif and Gene Structure Analysis of PEBP Family Members in Tomato

A phylogenetic tree containing only SLPEBP family genes with subfamilies divided into three groups was constructed by MEGA-X, consistent with the previous evolutionary tree results, and based on this tree, we performed conserved motif and gene structure analysis of SLPEBP gene family members (Figure 4).

In the gene structure analysis, six SLPEBP family genes contained four exons and two introns, accounting for 50% of the overall SLPEBP gene family members. Nine genes contained four exons, accounting for 75% of the family members. Seven genes contained two introns, accounting for 58.3% of the family members. Nine genes contained introns, accounting for 75% of the family members. The majority of SLPEBP gene family members contained four exons, and most of its members contained one or two introns. In the analysis of conserved motifs, 10 conserved motifs were found and named motifs 1–10. All genes except *SLPEBP3*, *SLPEBP6*, *SLPEBP8*, *SLPEBP10*, and *SLPEBP12* contained all 10 conserved motifs, and the 10 motifs were arranged in the same order. Only motifs 3 and 4 are shared by all 12 SLPEBP gene family members. The most complete conserved motif is found in the TFL1 subfamily, followed by the FT subfamily.

### 2.5. Prediction of the Three-Dimensional Structures of SLPEBP Proteins

The structure of the protein encoded by a gene is related to the function of the gene. Homology modeling is a method to construct the three-dimensional structure of a target protein by using the amino acid sequence of the protein as a template for the three-dimensional structure of a protein that has been experimentally validated. Using the homology modeling method, we predicted the 3D structures of 12 SLPEBP proteins. The 3D structures obtained by the homology modeling method for all 11 genes except *SLPEBP10* were evaluated as accurate by online software detection and evaluation. As shown in Figure 5, all 11 of these proteins contain a large β-fold, and 10 of them (the exception is *SLPEBP8*) contain two α-helices.

### 2.6. Prediction of Cis-Acting Elements of SLPEBP Family Members

Cis-acting elements are important for gene transcription, and in this study, a 2000 bp sequence upstream of each gene was used to predict cis-acting elements in the 12 genes of the SLPEBP family (Figure 6). A total of 36 cis-acting elements were detected and classified into six categories, developmental-associated elements (18), environmental stress-related elements (27), hormone-responsive elements (84), light-responsive elements (141), promoter-associated elements (1166), and site-binding-associated elements (13), for a total of 1448 predicted cis-acting elements. All SLPEBP genes except the *SLPEBP2* gene contained G-box. All family members except *SLPEBP4* contained Box 4. Among the cis-acting elements in the promoter region of the SLPEBP family members, the number of light-responsive elements (141) was the highest. The number of development-related elements was the lowest (18); *SLPEBP7* did not contain any development-related elements. Details of cis-acting elements in the promoter region of SLPEBP genes can be found in Appendix A.

### 2.7. Expression of SLPEBP Genes in Different Tissues and Organs of Tomato

The results of the analysis are shown in Figure 7. Four genes were significantly expressed in the buds, flowers, and roots, and two genes were significantly expressed in the leaves. Some genes (*SLPEBP3*, *SLPEBP12*) were expressed in several tissues, while some genes (*SLPEBP8*, *SLPEBP10*) were not found to be expressed in any of the tissues examined. Average FPKM value of SLPEBP genes transcriptome in different tissues of tomato can be found in Appendix A.

### 2.8. Analysis of qRT-PCR Gene Expression in the SLPEBP Gene Family

We determined the gene expression of all 12 genes, *SLPEBP1* to *SLPEBP12*, at five different periods of flowering, and the results are shown in Figure 8. Among them, the expression of *SLPEBP1* and *SLPEBP12* was extremely low and is therefore not shown in the figure. Among the 12 genes, *SLPEBP3*, *SLPEBP5*, *SLPEBP6*, *SLPEBP8*, *SLPEBP9*, and *SLPEBP10* were expressed at high levels during the flowering period and might be involved in flower opening. The other genes may be related to ovary formation and may play a regulatory role in tomato yield.

## 3. Discussion

The type of plant growth (determinate, indeterminate, and semi-determinate), the time of flowering, the number of flowers, and the development of flower organs are closely related to the yield of tomato. Currently, mechanized and automatic harvesting and processing are gradually replacing traditional manual work, further improving production efficiency and saving time and cost. To adapt to mechanized harvesting, it is necessary to study tomato plant types. Members of the PEBP gene family have obvious functions in plant growth type, flowering period, flower organ development, seed germination, root morphogenesis, and even abiotic stress resistance. Mutants of its genes in this family have been reported to cause changes in tomato flowering, flower morphology, plant growth type, root morphology, etc. However, no article has provided a comprehensive study and summary of the PEBP gene family in tomato to examine the common characteristics of its family members. We predicted 12 genes containing PEBP conserved domains through the tomato genome database, which is consistent with the results of Cao et al. [35] and Moreira et al. [30]. Through data analysis, studies found that the genes in this family are involved in transcriptional regulation; moreover, stress and signal transduction have strong homology and a large gene family [36,37,38], while the genes involved in basic functions have weak homology and a small gene family [39]. The SLPEBP gene family has only 12 members. It can be inferred that the gene function of the SLPEBP gene family is relatively basic. Using the same method, we predicted the PEBP gene family members of Arabidopsis, potato, and pepper, and carried out the same analysis. Seven members of the Arabidopsis PEBP gene family were found. The results of 10 potato PEBP gene family members are different from those of the 15 PEBP gene family members detected by Zhang et al., due to the different potato reference genomes [40]. The results of 11 pepper PEBP gene family members are slightly different from those of the 9 PEBP gene family members detected by Niu et al. due to different reference genomes and methods [41]. We found that the 12 genes of the SLPEBP gene family were unevenly distributed across seven tomato chromosomes (Figure 1), and all of them except *SLPEBP5* were distributed at the far end of the chromosome.

Gene replication events often occur in plants. In this way, plants can evolve with higher efficiency [42]. The members of the gene family are all from the same ancestor, and a group of genes that encode similar protein products with similar sequence and structure are formed by mutation, domestication, and selection through gene replication events or copies produced by whole-genome duplication (WGD) [43]. Gene duplication events can be divided into whole-genome duplications and single-gene duplications [44,45,46]. Repeated genes produced by whole-genome duplication are often lost or silenced after a period of time, and plants return to the diploid state quickly [47]. Single-gene duplication can occur in many ways, such as tandem duplication (TD), proximate duplication (PD), diffuse duplication (TRD), and separated duplication (DSD) [48]. More than 60% of the genes in the tomato genome are derived from gene replication events [45]. Intraspecific collinearity analysis showed that only the *SLPEBP1* and *SLPEBP9* genes had collinearity, indicating that SLPEBP gene family members had fewer copies, or that their copies were discarded more during evolution. At the same time, to further analyze the homologous relationships of SLPEBP gene family members in plants, we carried out interspecific collinearity analysis on tomato, potato, pepper, and Arabidopsis. The highest to lowest collinearity observed is potato (11), pepper (10), and Arabidopsis (4). This result is consistent with genetic distance [48]. *SLPEBP2* and *SLPEBP5* have no collinear relationship with the other three species and may be unique to tomato. *SLPEBP4*, *SLPEBP6*, and *SLPEBP9* and potato, pepper, and Arabidopsis all have a collinear relationship. These three genes may play an important role in the evolution of the PEBP gene family, or have irreplaceable functions. The phylogenetic tree of PEBP gene family members of tomato, Arabidopsis, pepper, and potato was constructed using MEGA-X. The evolutionary tree divides these members into three subgroups, namely TFL1, FT, and MFT (Figure 3). In the TFL1 branch, Arabidopsis *TFL1* (*AT5G03840*) and *FT* are antagonistic. *TFL1* has been shown to move in the apical meristem (SAM), regulate the floral transition, and regulate the expression of hundreds of genes [49]. Arabidopsis *ATC (AT2G27550)* can inhibit floral transition, and its expression is induced under short-day conditions [50]. Tomato *SP3C* (*SLPEBP4*) was found to be able to promote seed germination after being knocked out, and its overexpression can improve the tolerance of seedlings to water stress, and is related to root morphogenesis [30]. Puneli et al. found that *SP* (*SLPEBP9*) is homologous to *CEN* and *TFL1* [15]. The mutation of a single amino acid in the *SP* gene changed tomato from indeterminate to determinate growth. The *Solanum pennellii SP9D* (*SLPEBP11*) infiltration system may be associated with semi-determinate growth [29]. In the FT branch, Arabidopsis *FT* (*AT1G65480*) is a kind of floral integratorgene that plays a role in the floral transition process [51]. Arabidopsis *BFT* (*AT5G62040*) was found to play a role in the development of axillary inflorescence and regulate floral transition under high salt and drought conditions [52,53]. Arabidopsis *TSF* (*AT4G20370*) interacts with *FT*, but the direction of action is unknown at present [54]. Tomato *SP5G* (*SLPEBP7*) was found to be related to the photoperiod sensitivity of tomato. It is a kind of flowering replicator, and its variation enables the cultivated tomato to be planted in a wider area without being affected by the photoperiod [55]. Zhang et al. found that the photosensitive change in wild tomato from long-day plant to cultivated tomato was caused by the loss of 52 bp in the enhancer of the *SP5G* 3′ untranslated region [34]. Tomato *SP3D/SFT* (*SLPEBP5*) is an important gene controlling flower formation, responsible for encoding florigen and regulating the transformation of tomato from vegetative growth to reproductive growth [56]. When the *SFT* function loss allele is heterozygous, it can increase the yield by more than 60% and has strong heterosis [33]. By adjusting the balance between florigen and anti-florigen, the yield of tomato can be adjusted and optimized slightly to maximize the yield [57]. Potato *SP6A* (*PGSC0003DMT400060057*) controls the tuber vascular bundle transport signals and forms a complex with *IT1* to regulate the formation of potato tubers. The absence of its PEBP domain leads to the failure of *Solanum etuberosum* to produce potatoes [58]. In the MFT branch, the *MFT* (*AT1G18100*) gene of Arabidopsis was found to promote flowering and regulate the germination of Arabidopsis seeds through the ABA and GA signaling pathways [11,59]. Through collinearity and phylogenetic tree analysis, it was found that the function of the PEBP gene family is related to the synthesis of florigen, the regulation of floral transition, light sensitivity, plant growth and development, and some abiotic stress responses. In addition, the function of the PEBP gene family in different species varies greatly. It is obvious that Arabidopsis is a short-day plant, while tomato is a long-day plant. The response results of the PEBP gene family in these two plants are similar, but the regulatory modes are opposite.

In the analysis of the conserved motifs of the SLPEBP genes, only motifs 6 and 8 of the 10 conserved motifs were present in all 12 SLPEBP genes. All genes except for *SLPEBP3*, *SLPEBP6*, *SLPEBP8*, *SLPEBP10*, and *SLPEBP12* contain all 10 conserved motifs, and the 10 motifs are arranged in the same order. At present, there are no detailed reports on the functions of these five genes. Among the genes containing all 10 conserved motifs, *SLPEBP1* and *SLPEP2* have not been found to have specific functions. *SLPEBP3*, *SLPEBP6*, and *SLPEBP10* belong to the MFT subfamily. The FT and TFL1 subfamilies are believed to have evolved from the genes of this subfamily, which may be the reason why the conserved motifs of these three genes are incomplete. The prediction of the conserved motifs of the SLPEBP gene family provides a theoretical basis for gene classification and gene function prediction. The gene structure of the SLPEBP gene family is basically composed of four exons, with 75% of family members containing four exons, while there is loss of introns in the gene family, with the family members containing at most two introns. *SLPEBP8* has five exons, and *SLPEBP10* contains only two exons. *SLPEBP12* contains eight exons, and the gene structure is in good agreement with the conserved motif analysis, so it is likely that two genes were spliced together. The three-dimensional structure of proteins determines how they interact with other molecules, and determines their role in cells. The abundant three-dimensional structure of proteins allows them to perform a considerable number of functions in cells, and the three-dimensional structure of proteins can be inferred from their amino acid sequences [60]. From the three-dimensional structure prediction of proteins, we can see that the SLPEBP gene family is composed of a similar domain, among which *SLPEBP1*, *SLPEBP2*, and *SLPEBP4* have very similar three-dimensional structure prediction results, and they belong to the same subfamily. The predicted results are in line with the characteristics of a gene family protein, and all have similar conserved domains. In the process of protein transcription, the gene promoter and its cis-acting element play an important regulatory role. The core promoter of the gene encoding the protein usually contains a “TATA-box”. The cis-acting element and core promoter can become the binding sites of transcription and regulate the specific binding sites of the protein [61,62,63]. Through the analysis of cis-acting elements, it was found that all 12 members of the SLPEBP gene family contain the “TATA-box”. The analysis results of cis-acting elements can be divided into six categories: 18 development-related elements (8 kinds), 27 environmental stress-related elements (5 kinds), 84 hormone-response elements (9 kinds), 141 photo-response elements (20 kinds), 1166 promoter-related elements (4 kinds), and 13 site-binding related elements (4 kinds). Light-responsive elements were the most abundant in the SLPEBP gene family. In addition to cis-acting elements related to basic functions (promoter-related elements and site-binding related elements), the number of light-responsive elements is also the greatest, which proves that the functions of SLPEBP gene family members are related to light response, which is consistent with the results discussed earlier. In addition, SLPEBP gene family members are also involved in plant growth and development, abiotic stress response, hormone regulation, and other functions. This analysis result can also be verified in the previous discussion. After identifying the cis-acting elements in the promoter region, we found that the cis-acting elements were irregularly scattered in the promoter region (Figure 6B). The cis-acting elements distributed in different positions are conducive to regulating the functions of proteins under different environmental stimuli. Some members of the SLPEBP gene family, such as *SP9D*, *SP*, *SP5G*, and *SP3D*, have been shown to be expressed in many tissues and organs of tomato [29]. However, no study has been conducted to analyze the expression of all 12 SLPEBP genes in different tissues and organs of tomato. The expression and regulatory mechanisms of these 12 SLPEBP gene family members have been studied more clearly only for *SP*, *SFT*, and *SP5G*. By analysis of RNA-seq data, we found that some members of the SLPEBP gene family have tissue-specific expression (Figure 7). Among them, *SLPEBP1*, *SLPEBP2*, *SLPEBP9*, and *SLPEBP11* were specifically expressed in tomato roots, which may be related to root development or growth regulation, and interestingly, they all belong to the TFL1 subfamily. *SLPEBP4*, *SLPEBP6*, and *SLPEBP12* were specifically expressed in buds and flowers, which may be related to flower differentiation and development. *SLPEBP5* was specifically expressed in flowers, and *SLPEBP3* was specifically expressed in buds. *SLPEBP8* and *SLPEBP10* were not found to be expressed in any of the tissues and organs tested, and their expression may not be involved in the tested organs of tomato or they are not expressed in Heinz 1706. We selected five different stages of tomato from flowering to fruit setting for tissue-specific expression analysis, hoping to identify at which stage the SLPEBP gene family plays a role in flowering. *SLPEBP3*, *SLPEBP5*, *SLPEBP6*, *SLPEBP8*, *SLPEBP9*, and *SLPEBP10* have high expression in buds or incompletely opened flowers, and we speculate that they may be involved in the regulation of flowering in tomato. The other genes had higher variation in expression during flowering or fruit formation, so it is speculated that they may be involved in the regulation of ovary development.

## 4. Materials and Methods

### 4.1. Tomato Growth and Treatment

The cultivated tomato Ailsa Craig served as the experimental material. The plant material was planted in April 2022 in a solarium at Northeastern Agricultural University and sampled in October (126.916 E, 45.773 N). Tomato flower buds, incompletely opened flowers, completely opened flowers, and organs with 3 and 5 days of fruit set were collected to examine gene expression.

### 4.2. Total RNA Extraction and cDNA Synthesis

Total RNA was extracted from tomato materials using TaKaRa RNA extraction kit (Takara Bio, Beijing, China). cDNA synthesis and gDNA removal were performed using reagents produced by Vazyme (Vazyme, Nanjing, China). The obtained cDNA was stored at −80 °C.

### 4.3. Identification of Tomato PEBP Family Members

The genome sequence information of tomato, potato, pepper, and Arabidopsis was downloaded from the Ensemble database (http://plants.ensembl.org/index.html, accessed on 4 September 2021) for the identification of tomato PEBP family members and subsequent analysis. The conserved domain of PEBP (PF01161) was downloaded from Pfam (http://pfam.xfam.org/, accessed on 4 September 2021) [64]. HMMER3.0 (http://hmmer.org/, accessed on 4 September 2021) was used to retrieve tomato protein data and obtain possible tomato PEBP family members. Then, HMMER (https://www.ebi.ac.uk/Tools/hmmer/, accessed on 4 September 2021) and SMART (http://smart.embl.de/, accessed on 4 September 2021) online software were used to screen and confirm whether these genes contained conserved PEBP domains, and genes without complete domains were eliminated [65,66]. Pseudogenes were eliminated using REtrotransposed Gene EXPlorer [67]. ExPASy (https://web.expasy.org/protparam/, accessed on 4 September 2021) was used to predict the physicochemical properties of the SLPEBP, e.g., MW (kDa), theoretical pI, instability index, etc., and Psort (https://www.psort.org/, accessed on 4 September 2021) was used to predict the subcellular localization of the SLPEBP proteins [68,69].

### 4.4. Relationship between Chromosome Position and Replication of PEBP Family Genes

The chromosome position data, including chromosome length and gene starting and ending positions, were obtained from Solanaceae Genomics Network (SGN) (https://solgenomics.net/, accessed on 8 September 2021). A chromosome distribution map of the SLPEBP genes was drawn using TBtools [70]. Multiple collinear scanning toolkits (MCScanX) were used to analyze gene replication events, and the default values were adopted [71]. The collinear relationships within the SLPEBP gene family and the collinear relationships with genes in other species homologous to the SLPEBP gene family were plotted by Dual Systeny Plotter software [70].

### 4.5. Multiple Sequence Alignment and Phylogenetic Analysis

The PEBP family members in Arabidopsis, potato, and pepper were identified by the same method as that used to identify the PEBP family members in tomato, and seven PEBP family members in Arabidopsis, ten PEBP family members in potato, and eleven PEBP family members in pepper were identified by screening conserved structural domains using online software. A total of 7 PEBP family members from Arabidopsis, 10 PEBP family members from potato, 11 PEBP family members from pepper, and 12 PEBP family members from tomato were sequenced using MEGA-X, and the results were analyzed by phylogenetic analysis. The analysis was performed using neighbor-joining (NJ) and the protein evolution model JTT (Jones–Taylor–Thornton), as well as 1000 bootstrap replicates, and other default values.

### 4.6. Gene Structure and Conserved Motif Analysis

Gene Structure Display Server 2.0 (GSDS) (http://gsds.gao-lab.org/, accessed on 15 September 2021) was used to construct the gene structure map [72]. The conserved motifs in the SLPEBP family genes were predicted by MeMe (https://meme-suite.org/meme/tools/meme, accessed on 15 September 2021). The predicted number of motifs was 10, and the other was the default value.

### 4.7. Prediction of Three-Dimensional Protein Structure of Tomato PEBP

SWISS-MODEL (https://swissmodel.expasy.org/, accessed on 7 October 2021) was used to predict the three-dimensional structure of the protein by homology modeling [73,74,75,76,77]. A structure with sequence consistency greater than 30% was selected as the template, and the protein model was evaluated by SAVES v6.0 (https://saves.mbi.ucla.edu/, accessed on 7 October 2021) [78]. The template that passes three or more assessments was selected as the final template. Structural visualization was carried out by the 3D protein structure visualization software VMD (http://www.ks.uiuc.edu/Research/vmd, accessed on 7 October 2021) [79].

### 4.8. Analysis of Cis-Acting Elements of the SLPEBP Gene Family Members

The sequence of 2000 bp upstream of the transcription initiation site was extracted as the promoter sequence of the SLPEBP gene, and PlantCARE (http://bioinformatics.psb.ugent.be/webtools/plantcare/html/, accessed on 6 October 2021) was used to predict the potential cis-acting elements on the promoter sequence [80]. The prediction results were visualized, classified, and analyzed.

### 4.9. Analysis of SLPEBP Gene Expression in Different Tissues of Tomato

To investigate the differences in expression of the SLPEBP gene in different tissues of tomato, FPKM (fragment per million exons mapping) data for tomato variety Heinz 1706 with TFGD (http://ted.bti.cornell.edu/, accessed on 2 April 2023) accession number D004 were used [81]. Twelve SLPEBP genes were screened using these data and analyzed expression in buds; fully open flowers; leaves; roots; 1 cm, 2 cm, and 3 cm fruits; and mature green, breaker, and breaker +10 fruits of tomato cultivar Heinz 1706 [81]. To measure gene expression levels, the total number of FPKM values for each gene was calculated based on the length of the gene and the count of reads mapping to that gene. The data for each row were normalized and plotted according to the average FPKM value for each gene. The Euclidean distance between the genes was also analyzed and they were clustered. The R package pheatmap was used to generate the heatmap.

### 4.10. Analysis of SLPEBP Family Member Gene Expression

Specific primers were designed using NCBI (https://www.ncbi.nlm.nih.gov/, accessed on 4 September 2022) and synthesized by BGI Genomics Co., Ltd. Detailed primer sequence information can be found in Appendix A. AceQ qPCR SYBR Green Master Mix (Without ROX) reagent from Nanjing Vazyme Company was used, configured as a 20 µL system according to the instructions, and qRT-PCR analysis was performed using the German Analytik Jena quantitative PCR instrument qTOWER3G. Detailed procedures: Stage 1, 94 °C 2 min for 1 cycle; Stage 2, 94 °C 15 s, 57 °C 15 s, 72 °C 30 s for 40 cycles (scan); melting curve 60 to 90 °C, 10 s with ΔT 0.5 °C (scan). The *Actin* gene was used as the internal reference gene as the control for data standardization. The final calculation 2^−ΔΔ Ct^ was used to analyze the relative expression of genes. GraphPad Prism 9 was used to analyze the variance, and one-way ANOVA (and nonparametric or mixed) was used to analyze the significance.

## 5. Conclusions

In this study, 12 SLPEBP gene family genes were identified and analyzed. There is a collinear relationship between *SLPEBP1* and *SLPEBP9*. Eleven collinear relationships of tomato with potato, ten collinear relationships with pepper, and four collinear relationships with Arabidopsis were found. Through the analysis and construction of the evolutionary tree, the PEBP gene family was divided into three subfamilies, TFL1, FT, and MFT, and the characteristics of the subgroups were summarized. The gene structure and conserved motifs of the 12 genes were analyzed, and their general characteristics were summarized. The protein models of 11 members of the SLPEBP gene family were predicted. They all contain a similar conserved domain. Through the prediction of cis-acting elements, the function of SLPEBP gene family members was further analyzed. We speculate that SLPEBP gene family members have functions in plant growth and development, light response, hormone regulation, and some abiotic stress responses. Tissue-specific expression analysis showed changes in the expression of 10 members of the SLPEBP gene family in five different stages, and verified their role in plant growth and development. Six genes (*SLPEBP3*, *SLPEBP5*, *SLPEBP6*, *SLPEBP8*, *SLPEBP9*, and *SLPEBP10*) are hypothesized to be associated with tomato flowering and four (*SLPEBP2*, *SLPEBP3*, *SLPEBP7*, and *SLPEBP11*) may be associated with ovary development. The above analysis can provide research directions and ideas for future research on SLPEBP gene family members.

## Figures and Tables

**Figure 1 ijms-24-09185-f001:**
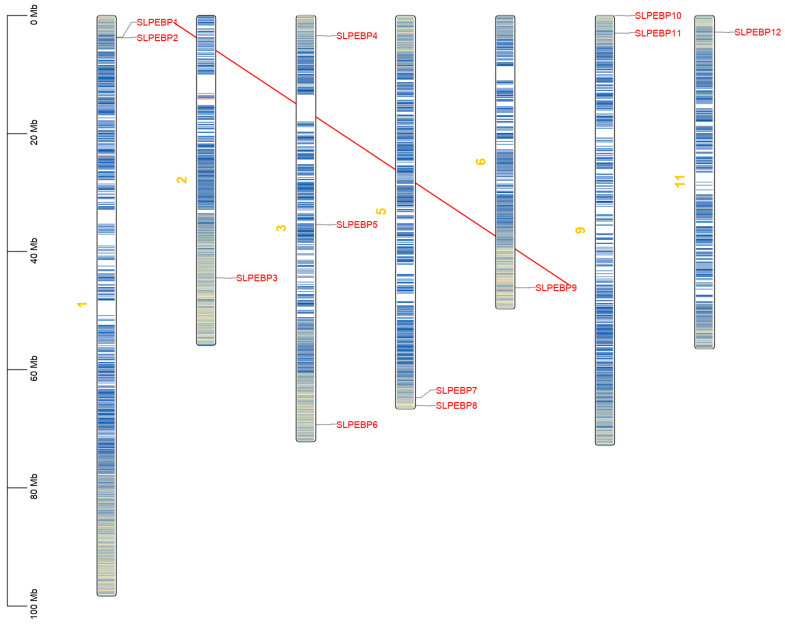
The position of SLPEBP gene family members on chromosomes and intraspecific collinearity. The scale on the left is used to estimate the length of chromosomes. The SLPEBP gene family members are numbered sequentially from 1 to 12 and marked in red. The yellow numbers represent chromosome numbers. Blue to red on the chromosome indicates gene density. The red line indicates the intraspecific collinearity of SLPEBP gene family members.

**Figure 2 ijms-24-09185-f002:**
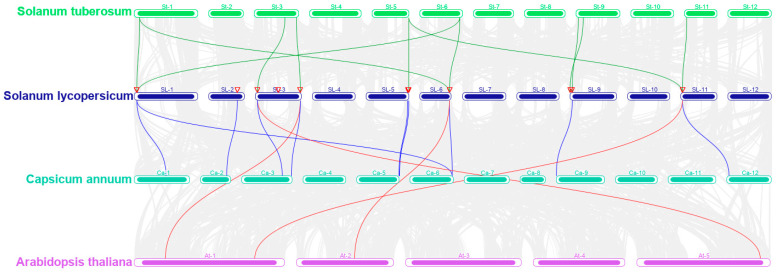
Interspecific collinearity relationship between SLPEBP gene family members and *Arabidopsis thaliana*, *Solanum tuberosum*, and *Capsicum annuum*. The chromosomes of *Solanum lycopersicum*, *Arabidopsis thaliana*, *Solanum tuberosum*, and *Capsicum annuum* are marked with different colors. The chromosome number is marked above the chromosome. The collinear relationship between the PEBP gene family members of different species and the SLPEBP gene family members is connected by different colored lines. Green lines, tomato with potato; blue lines, tomato with pepper; red lines, tomato with Arabidopsis. The red triangle represents the location of SLPEBP genes.

**Figure 3 ijms-24-09185-f003:**
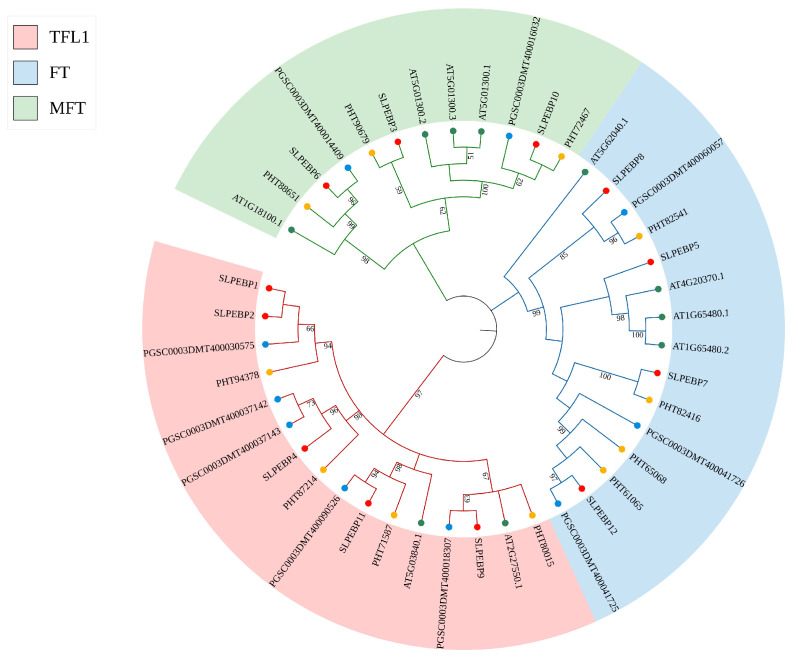
Phylogenetic tree: 12 SLPEBP genes, 7 ATPEBP genes, 10 STPEBP genes, and 11 CAPEBP genes. Different species are marked with solid dots of different colors. Red represents tomato, green represents Arabidopsis, blue represents potato, and yellow represents pepper. The three colors represent three different subfamilies of the PEBP gene family.

**Figure 4 ijms-24-09185-f004:**
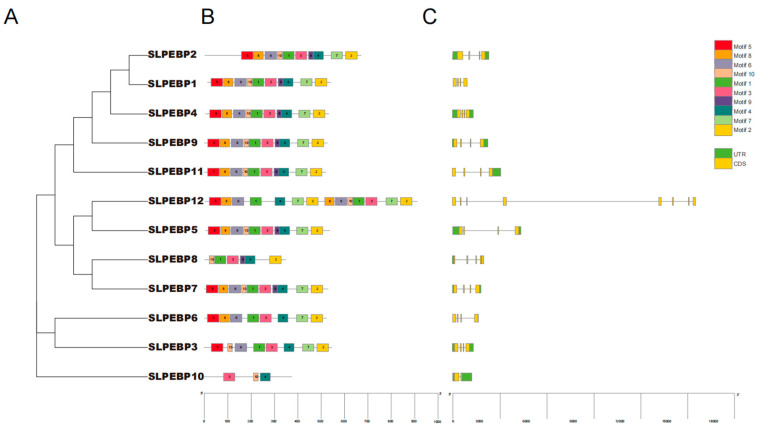
Gene structure and conserved motifs in SLPEBP gene family members. (**A**) Phylogenetic tree of SLPEBP gene family members. (**B**) SLPEBP conserved motif distribution order. Different motifs are marked with boxes of different colors. (**C**) Distribution of UTRs and CDSs of SLPEBP gene family members. Green represents UTRs and yellow represents CDSs. The scale at the bottom is used to compare the lengths of different genes and proteins.

**Figure 5 ijms-24-09185-f005:**
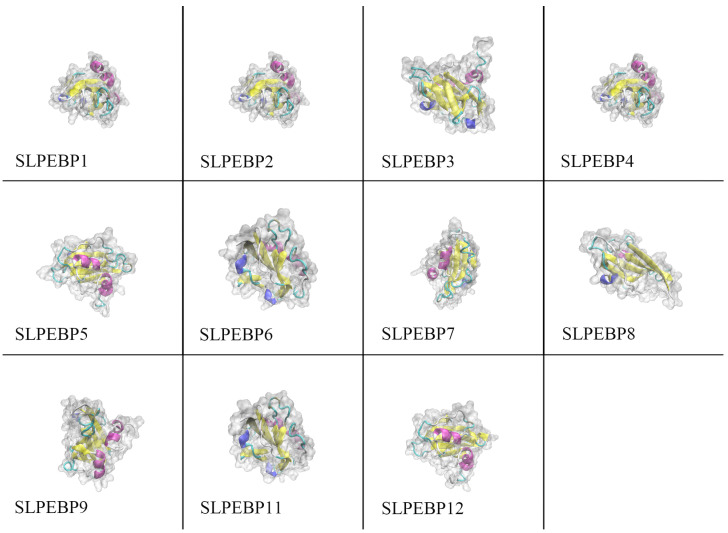
Three-dimensional structure model of SLPEBP family member proteins. Yellow represents β-fold, pink represents α-helix, cyan represents β-corner, and blue represents irregular curl. The translucent part shows the outer surface of the protein.

**Figure 6 ijms-24-09185-f006:**
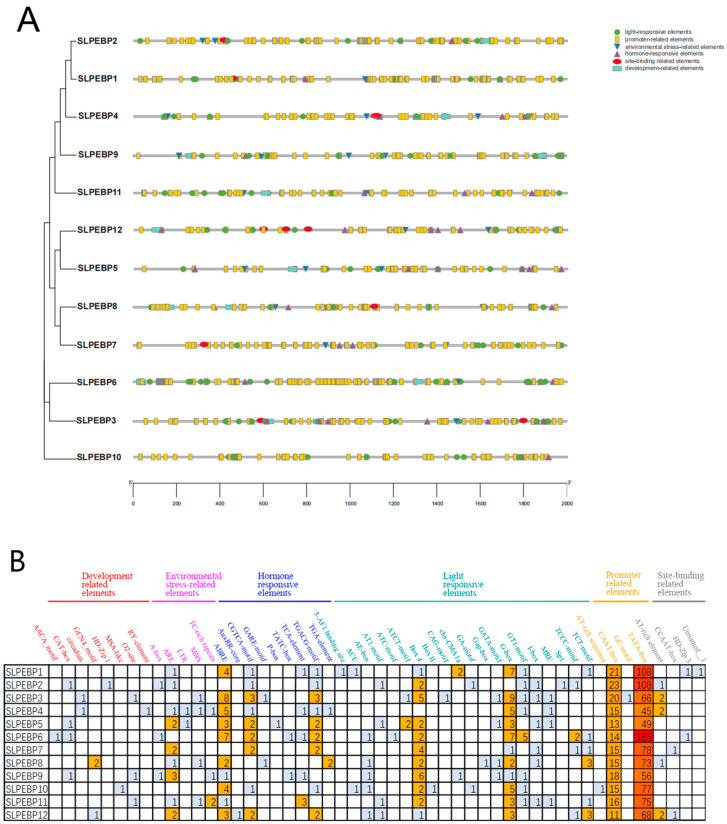
Prediction and analysis of cis-acting elements in the SLPEBP gene family promoter region (−2000 bp). (**A**) The distribution of cis-acting elements in the SLPEBP promoter region (−2000 bp). Different colors and shapes represent different types of cis-acting elements. The ruler at the bottom indicates the direction and length of the sequence. (**B**) Classification and statistics of cis-acting elements. Thirty-eight kinds of cis-acting elements were divided into six categories. The number in the grid represents the number of elements, and the color from blue to red represents the number of elements from fewer to more.

**Figure 7 ijms-24-09185-f007:**
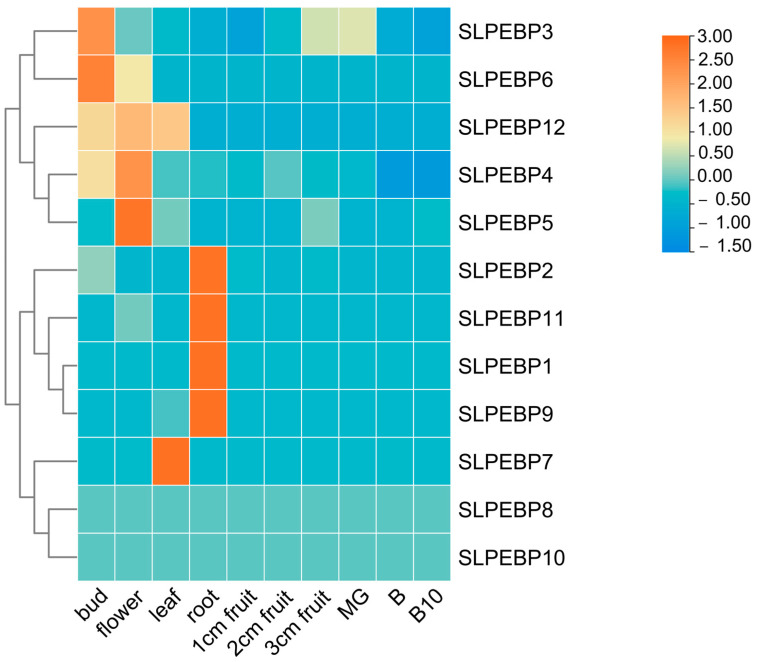
Expression of SLPEBP genes in different tissues. A heatmap was plotted after row normalization of the absolute fragment number per kilobase transcript (FPKM) values for each gene in different tomato organs. The dendrogram on the left shows the results of inter-gene clustering analysis.

**Figure 8 ijms-24-09185-f008:**
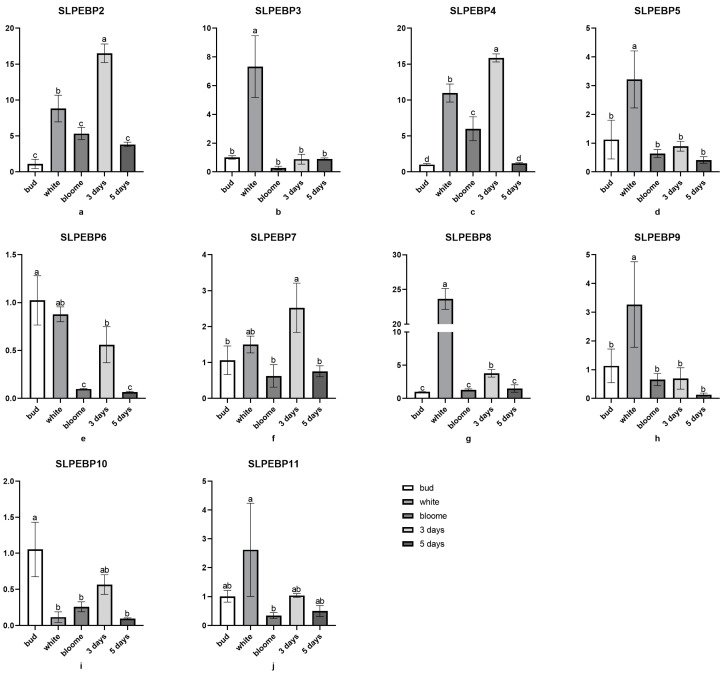
Analysis of gene expression of 10 SLPEBP gene family members in tomato materials at five different stages of flowering. Different shades of gray represent different flowering times. Different letters indicate significant differences, and the same letters represent no significant differences. The bars show the standard deviation.

**Table 1 ijms-24-09185-t001:** SLPEBP gene family protein properties table.

Gene Name	CDS Length (bp)	Chr	Position (5′–3′)	Protein Length (aa)	Protein Characteristics					Subcellular Location
					MW (kDa)	Theoretical pI	Instability Index	Aliphatic Index	GRAVY	
*SLPEBP1*	522	1	3,747,484–3,748,390	173	19.32	8.69	30.95	86.07	−0.221	Cyto
*SLPEBP2*	669	1	3,762,292–3,764,615	222	24.80	9.63	40.56	89.05	−0.218	Cyto
*SLPEBP3/SP2G*	546	2	44,460,468–44,461,817	181	20.12	6.05	37.06	72.71	−0.458	Cyto
*SLPEBP4/SP3C*	528	3	3,481,034–3,482,366	175	19.51	9.23	50.02	75.71	−0.362	Cyto
*SLPEBP5/SFT*	534	3	35,413,306–35,417,673	177	20.08	6.74	46.74	79.15	−0.388	Cyto
*SLPEBP6/SP3I*	522	3	69,283,358–69,285,003	173	19.15	8.60	39.32	82.14	−0.155	Cyto
*SLPEBP7* */SP5G*	528	5	64,738,354–64,740,162	175	19.54	5.26	30.19	85.09	−0.199	Cyto
*SLPEBP8/SP6A*	423	5	66,067,135–66,069,138	140	16.06	6.08	33.16	92.36	−0.172	Cyto
*SLPEBP9/SP*	528	6	46,113,204–46,115,474	175	19.93	8.72	50.26	75.03	−0.262	Cyto
*SLPEBP10*	480	9	38,884–40,108	159	17.83	5.54	28.46	84.59	−0.494	Cyto
*SLPEBP11/SP9D*	519	9	2,968,678–2,971,759	172	19.54	8.89	50.20	76.40	−0.377	Cyto
*SLPEBP12*	525	11	2,823,974–2,839,543	174	20.02	8.75	42.11	83.33	−0.370	Cyto

## Data Availability

The data presented in this study are openly available in the Tomato Functional Genomics Database (TFGD) (cornell.edu), reference number D004.

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
