# Peer review of "Genome-Wide Identification of PEBP Gene Family in Solanum lycopersicum"

_ijms, 2023, doi:10.3390/ijms24119185_

Round 1

Reviewer 1 Report

There are a number of misprints in "Genome-wide identification of PEBP gene family in Solanum lycopersicum":

Line 206: (figure6).
Line 236:

Also, a number of sentences are not built correctly, and they should be deleted or rewritten:

Line 68: Its fruit flavor is distinctive, may be consumed raw or cooked, and has a great financial and nutritional value.
Line 263: The SLPEBP gene family has only 12 members, with a small number of members.
Line 274: The replication events of plant genes often occur.
Line 352: The rich three-dimensional structure of protein enables it to exercise a considerable amount of functions in cells, and the three-dimensional structure of protein can be inferred from its amino acid sequence

How appropriate is the title of subsection 2.3. "Formatting of Mathematical Components"; if it is about building a phylogenetic tree?
Also not clear is the sentence beginning with line 379: "Some members of the SLPEBP gene family have been shown to be widely expressed in many tissues and organs of tomato", the authors write that they have found all the genes of this family. How then can results be obtained for these genes in a 20-year-old work?

Along with the above, the poor resolution of the drawings should be noted.

This publication, despite the significance of the experiment, needs significant revision.

Author Response

Reviewer #1:

1 There are a number of misprints in "Genome-wide identification of PEBP gene family in Solanum lycopersicum":

Line 206: (figure6).

Line 236:

Reply: Thank you for your correction. The two errors have been fixed. We have also checked the rest of the article to ensure that the same errors do not occur.

2 Also, a number of sentences are not built correctly, and they should be deleted or rewritten:

Line 68: Its fruit flavor is distinctive, may be consumed raw or cooked, and has a great financial and nutritional value.

Reply: Thank you, the incorrectness of this sentence has been removed.

3 Line 263: The SLPEBP gene family has only 12 members, with a small number of members.

Reply: Thank you for your correction, the structure of this sentence has been changed.

4 Line 274: The replication events of plant genes often occur.

Reply: We have rewritten this sentence and changed its structure.

5 Line 352: The rich three-dimensional structure of protein enables it to exercise a considerable amount of functions in cells, and the three-dimensional structure of protein can be inferred from its amino acid sequence

Reply: Thank you for your correction. The sentence did have an unclear expression and we have changed it to make sure the expression is correct.

6 How appropriate is the title of subsection 2.3. "Formatting of Mathematical Components"; if it is about building a phylogenetic tree?

Reply: Thank you very much for your careful examination. This was a very serious mistake in our writing process. We have changed the title of 2.3. It has been changed to "Phylogenetic relationship analysis".

7 Also not clear is the sentence beginning with line 379: "Some members of the SLPEBP gene family have been shown to be widely expressed in many tissues and organs of tomato", the authors write that they have found all the genes of this family. How then can results be obtained for these genes in a 20-year-old work?

Reply: Thank you very much for your question. It is true that this paragraph was not clear. We have changed this paragraph and added some content.

We did screen out 12 SLPEBP genes by bioinformatics approach. But unfortunately, we have to admit that the bioinformatics approach to analysis has some limitations. The tomato genome database is more complete, so the results are more reliable. For other horticultural crops that do not have better sequencing results, there are significant differences using different databases. However, we cannot completely conclude that these 12 genes are all members of the SLPEBP gene family, and perhaps this result will be overturned as the tomato genome becomes better sequenced.

It is also unfortunate that only three of these 12 genes, SP, SFT, and SP5G, have been studied in detail. the study of SP3C currently rests on function, while SP9D has only been briefly mentioned in some articles. Details about the expression levels of SP9D, SP, SP5G, and SP3D in different tissues can be found in the reference [30].

8 Along with the above, the poor resolution of the drawings should be noted.

Reply: Thank you for the heads up. We have rechecked the resolution of the images and changed the size of the images to make them more readable.

9 This publication, despite the significance of the experiment, needs significant revision.

Reply: Thank you for acknowledging the content of our research. We made lots of modifications in the revised manuscript, and the manuscript has been re-polished before submission.

Reviewer 2 Report

I think the journal choice is not quite appropriate as this paper nearly entirely bioinformatical.  Punctuation: space between the text and reference, no space after the reference and the dot, like this “ … word [12].”

 The text continues here.” – what does this mean?

 “sis thaliana ,Solanum tuberosum ,Capsicum annuum.” – proper punctuation, the coma should be right after the word

“, SLPEBP5,SLPEBP7,SLPEBP8,SLPEBP12” – punctuation between words

“We determined the gene expression of SLPEBP1 to SLPEBP12, 12 genes at different 5 periods of flowering, and the results are shown in Fig.” – “at five different periods”, insert Figure number.

“The vertical line 244 shows the standard deviation.” – “bars” show standard deviation

“Different gray levels represent different periods.” – different shades of gray represent different flowering time

Insert “different letters indicate significance”

“The plant type of tomato (deterministic growth and indeterminate growth),” – this is not “plant type”, it is type of plant growth, “determinate”, “indeterminate” and “semideterminate”

“…the number of flowering…” – the number of flowers

In Methods: “Twelve SLPEBP genes were screened and analyzed expression in buds, fully open flowers, leaves, roots, 1 cm, 2 cm, and 3 cm fruits, mature green, breaker, and breaker+10 fruits of tomato cultivar Heinz 1706.” – please explain that the data are extracted from a different paper/database

Primer sequences and parameters of the PCR should be indicated

Why did not you study the expression in different types of tomato, with determinate and indeterminate growth?

 “We speculate that SLPEBP gene family members have functions in plant growth and development, light response, hormone regulation and some abiotic stress response” – this could been easily tested

Grammar and punctuation must be significantly improved

Author Response

Reviewer #2:

1 I think the journal choice is not quite appropriate as this paper nearly entirely bioinformatical.

Reply:Thank you for your reminder and advice. We have carefully reviewed the types of journals and journal content published in the International Journal of Molecular Sciences again and found that the journal has received some bioinformatics-related content.

For example:

Endogenous Retrovirus RNA Expression Differences between Race, Stage and HPV Status Offer Improved Prognostication among Women with Cervical Cancer

Utility of High-Sensitivity Modified Glasgow Prognostic Score in Cancer Prognosis: A Systemic Review and Meta-Analysis

The Potential Role of Complement System in the Progression of Ovarian Clear Cell Carcinoma Inferred from the Gene Ontology-Based Immunofunctionome Analysis

This article is mainly a bioinformatic analysis, but is intended to provide a reference and direction for future molecular biology research on the PEBP gene family in tomato. 

2 Punctuation: space between the text and reference, no space after the reference and the dot, like this “ … word [12].”

Reply: Thank you for your correction. The sentences within the article that had the same problem were also corrected together.

3 “The text continues here.” – what does this mean?

Reply: We apologize that this was a mistake in the writing process. The phrase has been deleted. We have also checked the rest of the article to make sure the same error does not happen.

4 “sis thaliana ,Solanum tuberosum ,Capsicum annuum.” – proper punctuation, the coma should be right after the word

Reply: Thank you very much for such a detailed and patient check. We have corrected the punctuation between the words. Modified to "Arabidopsis thaliana, Solanum tuberosum, Capsicum annuum." Also, we have fixed similar problems in the article.

5 “, SLPEBP5,SLPEBP7,SLPEBP8,SLPEBP12” – punctuation between words

Reply: Thank you for the reminder. We have corrected the punctuation between the words.

6 “We determined the gene expression of SLPEBP1 to SLPEBP12, 12 genes at different 5 periods of flowering, and the results are shown in Fig.” – “at five different periods”, insert Figure number.

Reply: Thank you for your reminder. We have added figure numbers for this section. You can see the relevant content in Figure 8.

7 “The vertical line 244 shows the standard deviation.” – “bars” show standard deviation

Reply: Thank you for your correction. We have changed the inaccurate terminology.

8 “Different gray levels represent different periods.” – different shades of gray represent different flowering time Insert “different letters indicate significance”

Reply: Thank you for the corrections you provided. We have changed the expressions and added sentences explaining the meaning of the different letters.

9 “The plant type of tomato (deterministic growth and indeterminate growth),” – this is not “plant type”, it is type of plant growth, “determinate”, “indeterminate” and “semideterminate”

Reply: Thank you for the correction on the technical vocabulary. The inaccurate description has been changed.

10 “…the number of flowering…” – the number of flowers

Reply: The inaccurate description has been changed.

11 In Methods: “Twelve SLPEBP genes were screened and analyzed expression in buds, fully open flowers, leaves, roots, 1 cm, 2 cm, and 3 cm fruits, mature green, breaker, and breaker+10 fruits of tomato cultivar Heinz 1706.” – please explain that the data are extracted from a different paper/database

Reply: Thank you for your reminder. We have marked and explained the source of the data and cited the paper.

12 Primer sequences and parameters of the PCR should be indicated

Reply: Thanks again for the reminder. This was indeed a big oversight. We have added to this section of the PCR program. Details of the primers can be found in Table S4.

13 Why did not you study the expression in different types of tomato, with determinate and indeterminate growth?

Reply: Thank you for asking this question.

The vast majority of determinate growth types of tomatoes currently available are due to alterations in known SP genes. Therefore we judged that comparing the two types of tomatoes, the larger difference would be in the SP gene, which is not consistent with our experimental purpose.

Of course, we will consider comparing these two different growth types of tomatoes in a subsequent, more in-depth study of a particular SLPEBP gene family member.

14 “We speculate that SLPEBP gene family members have functions in plant growth and development, light response, hormone regulation and some abiotic stress response” – this could been easily tested

Reply: Thank you for your comments.

Among these 12 gene families, SFT, SP, SP5G, and SP3C, which have recently been validated, are the most well studied, and it is clear that their functions have been thoroughly studied, and it is not meaningful to validate their functions again. The other members of the SLPEBP gene family have hardly been reported, and there is no simple and efficient way to determine whether they are functional, which requires more comprehensive studies.

Round 2

Reviewer 1 Report

I am completely satisfied with the changes made.

Reviewer 2 Report

Most of my comments were addressed

Some minor changes are still needed.